# MULTI-AGENT IMITATION LEARNING WITH COPULAS

## ABSTRACT

Multi-agent imitation learning aims to train multiple agents to perform tasks from demonstrations by learning a mapping between observations and actions, which is essential for understanding physical, social, and team-play systems. However, most existing works on modeling multi-agent interactions typically assume that agents make independent decisions based on their observations, ignoring the complex dependence among agents. In this paper, we propose to use copula, a powerful statistical tool for capturing dependence among random variables, to explicitly model the correlation and coordination in multi-agent systems. Our proposed model is able to separately learn marginals that capture the local behavioral patterns of each individual agent, as well as a copula function that solely and fully captures the dependence structure among agents. Extensive experiments on synthetic and real-world datasets show that our model outperforms state-of-the-art baselines across various scenarios in the action prediction task, and is able to generate new trajectories close to expert demonstrations.

## 1 INTRODUCTION

Recent years have witnessed great success of reinforcement learning (RL) for single-agent sequential decision making tasks. As many real-world applications (e.g., multi-player games (Silver et al., 2017; Brown & Sandholm, 2019) and traffic light control (Chu et al., 2019)) involve the participation of multiple agents, multi-agent reinforcement learning (MARL) has gained more and more attention. However, a key limitation of RL and MARL is the difficulty of designing suitable reward functions for complex tasks with implicit goals (e.g., dialogue systems) (Russell, 1998; Ng et al., 2000; Fu et al., 2017; Song et al., 2018). Indeed, hand-tuning reward functions to induce desired behaviors becomes especially challenging in multi-agent systems, since different agents may have completely different goals and state-action representations (Yu et al., 2019).

Imitation learning provides an alternative approach to directly programming agents by taking advantage of expert demonstrations on how a task should be solved. Although appealing, most prior works on multi-agent imitation learning typically assume agents make independent decisions after observing a state (i.e., mean-field factorization of the joint policy) (Zhan et al., 2018; Le et al., 2017; Song et al., 2018; Yu et al., 2019), ignoring the potentially complex dependencies that exist among agents. Recently, Tian et al. (2019) and Liu et al. (2020) proposed to implement correlated policies with opponent modeling, which incurs unnecessary modeling cost and redundancy, while still lacking coordination during execution.

Compared to the single-agent setting, one major and fundamental challenge in multi-agent learning is how to model the dependence among multiple agents in an effective and scalable way. Inspired by probability theory and statistical dependence modeling, in this work, we propose to use copulas (Sklar, 1959b; Nelsen, 2007; Joe, 2014) to model multi-agent behavioral patterns. Copulas are powerful statistical tools to describe the dependence among random variables, which have been widely used in quantitative finance for risk measurement and portfolio optimization (Bouyé et al., 2000). Using a copulas-based multi-agent policy enables us to separately learn marginals that capture the local behavioral patterns of each individual agent and a copula function that only and fully captures the dependence structure among the agents. Such a factorization is capable of modeling arbitrarily complex joint policy and leads to *interpretable, efficient and scalable* multi-agent imitation learning. As a motivating example (see Figure 1), suppose there are two agents, each with one-dimensional action space. In Figure 1a, although two joint policies are quite different, they actually share the same copula (dependence structure) and one marginal. Our proposed copula-based policy is capable

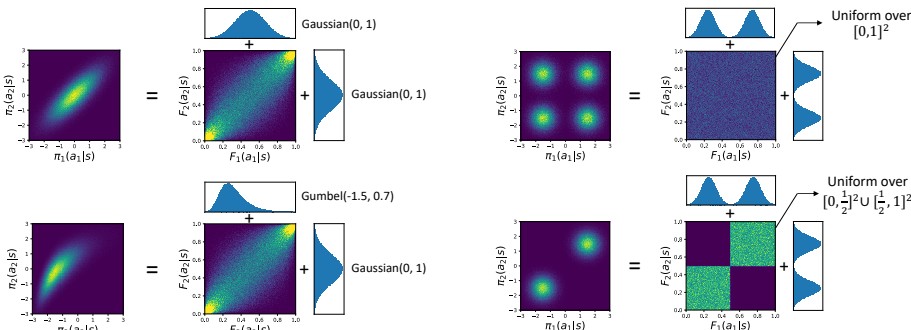

(a) Same copula but different marginals  (b) Same marginals but different copulas

Figure 1: In each subfigure, the left part visualizes the joint policy $\boldsymbol{\pi}(a_1, a_2|s)$ on the joint action space $[-3, 3]^2$ and the right part shows the corresponding marginal policies (e.g., $\pi_1(a_1|s) = \int_{a_2} \boldsymbol{\pi}(a_1, a_2|s)\mathrm{d}a_2$) as well as the copula $c(F_1(a_1|s), F_2(a_2|s))$ on the unit cube. Here $F_i$ is the cumulative distribution function of the marginal $\pi_i(a_i|s)$ and $u_i := F_i(a_i|s)$ is the uniformly distributed random variable obtained by probability integral transform with $F_i$. More details and definitions can be found in Section 3.2.

of capturing such information and more importantly, we may leverage such information to develop efficient algorithms for such transfer learning scenarios. For example, when we want to model team-play in a soccer game and one player is replaced by his/her substitute while the dependence among different roles are basically the same regardless of players, we can immediately obtain a new joint policy by switching in the new player's marginal while keeping the copula and other marginals unchanged. On the other hand, as shown in Figure 1b, two different joint policies may share the same marginals while having different copulas, which implies that the mean-field policy in previous works (only modeling marginal policies and making independent decisions) cannot differentiate these two scenarios to achieve coordination correctly.

Towards this end, in this paper, we propose a copula-based multi-agent imitation learning algorithm, which is interpretable, efficient and scalable for modeling complex multi-agent interactions. Extensive experimental results on synthetic and real-world datasets show that our proposed method outperforms state-of-the-art multi-agent imitation learning methods in various scenarios and generates multi-agent trajectories close to expert demonstrations.

## 2 PRELIMINARIES

In this work, we consider the problem of multi-agent imitation learning under the framework of Markov games (Littman, 1994), which generalize Markov Decision Processes to multi-agent settings, where $N$ agents are interacting with each other. Specifically, in a Markov game, $\mathcal{S}$ is the common state space, $\mathcal{A}_i$ is the action space for agent $i \in \{1, \ldots, N\}$, $\eta \in \mathcal{P}(\mathcal{S})$ is the initial state distribution and $P : \mathcal{S} \times \mathcal{A}_1 \times \ldots \times \mathcal{A}_N \to \mathcal{P}(\mathcal{S})$ is the state transition distribution of the environment that the agents are interacting with. Here $\mathcal{P}(\mathcal{S})$ denotes the set of probability distributions over state space $\mathcal{S}$. Suppose at time $t$, agents observe $s[t] \in \mathcal{S}$ and take actions $\boldsymbol{a}[t] := (a_1[t], \ldots, a_N[t]) \in \mathcal{A}_1 \times \ldots \times \mathcal{A}_N$, the agents will observe state $s[t + 1] \in \mathcal{S}$ at time $t + 1$ with probability $P(s[t + 1]|s[t], a_1[t], \ldots, a_N[t])$. In this process, the agents select the joint action $\boldsymbol{a}[t]$ by sampling from a stochastic joint policy $\boldsymbol{\pi} : \mathcal{S} \to \mathcal{P}(\mathcal{A}_1 \times \ldots \times \mathcal{A}_N)$. In the following, we will use subscript $-i$ to denote all agents except for agent $i$. For example, $(a_i, \boldsymbol{a}_{-i})$ represents the actions of all agents; $\pi_i(a_i|s)$ and $\pi_i(a_i|s, \boldsymbol{a}_{-i})$ represent the marginal and conditional policy of agent $i$ induced by the joint policy $\boldsymbol{\pi}(\boldsymbol{a}|s)$ (through marginalization and Bayes's rule respectively).

We consider the following off-line imitation learning problem. Suppose we have access to a set of demonstrations $\mathcal{D} = \{\tau^j\}_{j=1}^M$ provided by some expert policy $\boldsymbol{\pi}^E(\boldsymbol{a}|s)$, where each expert trajectory $\tau^j = \{(s^j[t], \boldsymbol{a}^j[t])\}_{t=1}^T$ is collected by the following sampling process:

$$s^1 \sim \eta(s), \boldsymbol{a}[t] \sim \boldsymbol{\pi}^E(\boldsymbol{a}|s[t]), s[t + 1] \sim P(s|s[t], \boldsymbol{a}[t]) \quad \text{for } t \in \{1, \ldots, T\}.$$

The goal is to learn a parametric joint policy $\boldsymbol{\pi}^\theta$ to approximate the expert policy $\boldsymbol{\pi}^E$ such that we can do downstream inferences (e.g., action prediction and trajectory generation). The learning problem is off-line as we cannot ask for additional interactions with the expert policy or the environment during training, and the reward is also unknown.

## 3 MODELING MULTI-AGENT INTERACTIONS WITH COPULAS

### 3.1 MOTIVATION

Many modeling methods for multi-agent learning tasks employ a simplifying mean-field assumption that the agents make independent action choices after observing a state (Albrecht & Stone, 2018; Song et al., 2018; Yu et al., 2019), which means the joint policy can be factorized as follows:

$$\boldsymbol{\pi}(a_1, \ldots, a_N|s) = \prod_{i=1}^{N} \pi_i(a_i|s) \tag{1}$$

Such a mean-field assumption essentially allows for independent construction of each agent's policy. For example, multi-agent behavior cloning by maximum likelihood estimation is now equivalent to performing $N$ single-agent behavior cloning tasks:

$$\max_{\boldsymbol{\pi}} \mathbb{E}_{(s,\boldsymbol{a})\sim\rho_{\boldsymbol{\pi}_E}}[\log \boldsymbol{\pi}(\boldsymbol{a}|s)] = \sum_{i=1}^{N} \max_{\pi_i} \mathbb{E}_{(s,a_i)\sim\rho_{\boldsymbol{\pi}_E},i}[\log \pi_i(a_i|s)] \tag{2}$$

where the occupancy measure $\rho_{\boldsymbol{\pi}} : \mathcal{S} \times \mathcal{A}_1 \times \ldots \times \mathcal{A}_N \to \mathbb{R}$ denotes the state action distribution encountered when navigating the environment using the joint policy $\boldsymbol{\pi}$ (Syed et al., 2008; Puterman, 2014) and $\rho_{\boldsymbol{\pi},i}$ is the corresponding marginal occupancy measure.

However, when the expert agents are making correlated action choices (e.g., due to joint plan and communication in a soccer game), such a simplifying modeling choice is not able to capture the rich dependency structure and coordination among agent actions. To address this issue, recent works (Tian et al., 2019; Liu et al., 2020) propose to use a different factorization of the joint policy such that the dependency among $N$ agents can be preserved:

$$\boldsymbol{\pi}(a_i, \boldsymbol{a}_{-i}|s) = \pi_i(a_i|s, \boldsymbol{a}_{-i})\boldsymbol{\pi}_{-i}(\boldsymbol{a}_{-i}|s) \quad \text{for } i \in \{1, \ldots, N\}. \tag{3}$$

Although such a factorization is general and captures the dependency among multi-agent interactions, several issues still remain. First, the modeling cost is increased significantly, because now we need to learn $N$ different and complicated opponent policies $\boldsymbol{\pi}_{-i}(\boldsymbol{a}_{-i}|s)$ as well as $N$ different marginal conditional policies $\pi_i(a_i|s, \boldsymbol{a}_{-i})$, each with a deep neural network. It should be noted that there are many redundancies in such a modeling choice. Specifically, suppose there are $N$ agents and $N > 3$, for agent 1 and $N$, we need to learn opponent policies $\boldsymbol{\pi}_{-1}(a_2, \ldots, a_N|s)$ and $\boldsymbol{\pi}_{-N}(a_1, \ldots, a_{N-1}|s)$ respectively. These are potentially high dimensional and might require flexible function approximations. However, the dependency structure among agent 2 to agent $N - 1$ are modeled in both $\boldsymbol{\pi}_{-1}$ and $\boldsymbol{\pi}_{-N}$, which incurs unnecessary modeling cost. Second, when executing the policy, each agent $i$ makes decisions through its marginal policy $\pi_i(a_i|s) = \mathbb{E}_{\boldsymbol{\pi}_{-i}(\boldsymbol{a}_{-i}|s)}(a_i|s, \boldsymbol{a}_{-i})$ by first sampling $\boldsymbol{a}_{-i}$ from its opponent policy $\boldsymbol{\pi}_{-i}$ then sampling its action $a_i$ from $\pi_i(\cdot|s, \boldsymbol{a}_{-i})$. Since each agent is performing such decision process independently, coordination among agents are still impossible due to sampling randomness. Moreover, a set of independently learned conditional distributions are not necessarily consistent with each other (i.e., induced by the same joint policy) (Yu et al., 2019).

In this work, to address above challenges, we draw inspiration from probability theory and propose to use copulas, a statistical tool for describing the dependency structure between random variables, to model the complicated multi-agent interactions in a scalable and efficient way.

### 3.2 COPULAS

When the components of a multivariate random variable $\boldsymbol{x} = (x_1, \ldots, x_N)$ are jointly independent, the density of $\boldsymbol{x}$ can be written as:

$$p(\boldsymbol{x}) = \prod_{i=1}^{N} p(x_i) \tag{4}$$

When the components are not independent, this equality does not hold any more as the dependencies among $x_1, \ldots, x_N$ can not be captured by the marginals $p(x_i)$. However, the differences can be corrected by multiplying the right hand side of Equation (4) with a function that *only and fully* describes the dependency. Such a function is called a copula (Nelsen, 2007), a multivariate distribution function on the unit hyper-cube with uniform marginals.

Intuitively, let us consider a random variable $x_i$ with continuous cumulative distribution function $F_i$. Applying *probability integral transform* gives us a random variable $u_i = F_i(x_i)$, which has standard uniform distribution. Thus one can use this property to separate the information in marginals from the dependency structures among $x_1, \ldots, x_N$ by first projecting each marginal onto one axis of the hyper-cube and then capture the pure dependency with a distribution on the unit hyper-cube.

Formally, a copula is the joint distribution of random variables $u_1, \ldots, u_N$, each of which is marginally uniformly distributed on the interval $[0, 1]$. Furthermore, we introduce the following theorem that provides the theoretical foundations of copulas:

**Theorem 1** ((Sklar, 1959a)). *Suppose the multivariate random variable $(x_1, \ldots, x_N)$ has marginal cumulative distribution functions $F_1, \ldots, F_N$ and joint cumulative distribution function $F$, then there exists a unique copula $C : [0, 1]^N \to [0, 1]$ such that:*

$$F(x_1, \ldots, x_N) = C\big(F_1(x_1), \ldots, F_N(x_N)\big) \tag{5}$$

*When the multivariate distribution has a joint density $f$ and marginal densities $f_1, \ldots, f_N$, we have:*

$$f(x_1, \ldots, x_N) = \prod_{i=1}^{N} f_i(x_i) \cdot c\big(F_1(x_1), \ldots, F_N(x_N)\big) \tag{6}$$

*where $c$ is the probability density function of the copula. The converse is also true. Given a copula $C$ and marginals $F_i(x_i)$, then $C\big(F_1(x_1), \ldots, F_N(x_N)\big) = F(x_1, \ldots, x_N)$ is a $N$-dimensional cumulative distribution function with marginal distributions $F_i(x_i)$.*

Theorem 1 states that every multivariate cumulative distribution function $F(x_1, \ldots, x_N)$ can be expressed in terms of its marginals $F_i(x_i)$ and a copula $C\big(F_1(x_1), \ldots, F_N(x_N)\big)$. Comparing Eq. (4) with Eq. (6), we can see that a copula function encoding correlations between random variables can be used to correct the mean-field approximation for arbitrarily complex distribution.

### 3.3 Multi-Agent Imitation Learning with Copula-based Policies

A central question in multi-agent imitation learning is how to model the dependency structure among agent decisions properly. As discussed above, the framework of copulas provides a mechanism to decouple the marginal policies (individual behavioral patterns) from the dependency left in the joint policy after removing the information in marginals. In this work, we advocate copula-based policy for multi-agent learning because copulas offer some unique and desirable properties in multi-agent scenarios. For example, suppose we want to model the interactions among players in a sports game. Using copula-based policy, we will obtain marginal policies for each individual player as well as dependencies among different roles (e.g., forwards and midfielders in soccer). Such a multi-agent learning framework has the following advantages:

- **Interpretable**. The learned copula density can be easily visualized to intuitively analyze the correlation among agent actions.

- **Scalable**. When the marginal policy of agents changes but the dependency among different agents remain the same (e.g., in a soccer game, one player is replaced by his/her substitute, but the dependence among different roles are basically the same regardless of players), we can obtain a new joint policy efficiently by switching in the new agent's marginal while keeping the copula and other marginals unchanged.

- **Succinct**. The copula-based factorization of the joint policy avoids the redundancy in previous opponent modeling approaches (Tian et al., 2019; Liu et al., 2020) (as discussed in Section 3.1) by separately learning marginals and a copula.

---

**Algorithm 1:** Training procedure

---

**Input:** The number of trajectories $M$, the length of trajectory $T$, the number of agents $N$, demonstrations
$\mathcal{D} = \{\tau^i\}_{i=1}^M$, where each trajectory $\tau^i = \{(s^i[t], \boldsymbol{a}^i[t])\}_{t=1}^T$
**Output:** Marginal action distribution MLP $MLP_{marginal}$, state-dependent copula MLP $MLP_{copula}$ or
state-independent copula density $c(\cdot)$

    // Learning marginals
1  **while** $MLP_{marginal}$ *not converge* **do**
2     **for** *each state-action pair* $(s, (a_1, \cdots, a_N))$ **do**
3         Calculate the conditional marginal action distributions for all agent:
           $\{f_j(\cdot|s)\}_{j=1}^N \leftarrow MLP_{marginal}(s)$
4         **for** *agent* $j = 1, \cdots, N$ **do**
5            Calculate the likelihood of the observed action $a_j$: $f_j(a_j|s)$
6            Maximize $f_j(a_j|s)$ by optimizing $MLP_{marginal}$ using gradient descent

    // Learning copula
7  **while** $MLP_{copula}$ *or* $c(\cdot)$ *not converge* **do**
8     **for** *each state-action pair* $(s, (a_1, \cdots, a_N))$ **do**
9         $\{f_j(\cdot|s)\}_{j=1}^N \leftarrow MLP_{marginal}(s)$
10       **for** *agent* $j = 1, \cdots, N$ **do**
11          $F_j(\cdot|s) \leftarrow$ the CDF of $f_j(\cdot|s)$
12          Transform the agent action $a_j$ to uniformly distributed value $u_j \leftarrow F_j(a_j|s)$
13       Obtain $u = (u_1, \cdots, u_N)$ in the unit hyper-cube $[0, 1]^N$
14       **if** *copula is set as state-dependent* **then**
15          Calculate the state-dependent copula density $c(\cdot|s) \leftarrow MLP_{copula}(s)$
16          Calculate the likelihood of $u$: $c(u|s)$
17          Optimize $MLP_{copula}$ by maximizing $\log c(u|s)$ using gradient descent
18       **else**
19          Calculate the likelihood of $u$: $c(u)$
20          Optimize parameters of $c(\cdot)$ using maximum likelihood or non-parametric methods

21  **return** $MLP_{marginal}$, $MLP_{copula}$ or $c(\cdot)$

---

**Learning.** In this section, we discuss how to learn a copula-based policy from a set of expert demonstrations. Under the framework of Markov games and copulas, we factorize the parametric joint policy as:

$$\boldsymbol{\pi}(a_1, \ldots, a_N|s; \boldsymbol{\theta}) = \prod_{i=1}^N \pi_i(a_i|s; \theta_i) \cdot c\big(F_1(a_1|s; \theta_1), \ldots, F_N(a_N|s; \theta_N)|s; \theta_c\big) \quad (7)$$

where $\pi_i(a_i|s; \theta_i)$ is the marginal policy of agent $i$ with parameters $\theta_i$ and $F_i$ is the corresponding cumulative distribution function; the function $c$ (parameterized by $\theta_c$) is the density of the copula on the transformed actions $u_i = F_i(a_i|s; \theta_i)$ obtained by processing original actions with probability integral transform.

The training algorithm of our proposed method is presented as Algorithm 1. Given a set of expert demonstrations $\mathcal{D}$, our goal is to learn marginal actions of agents and their copula function. Our approach consists of two steps.[1] We first learn marginal action distributions of each agent given their current state (lines 1-6). This is achieved by training $MLP_{marginal}$ that takes as input a state $s$ and output the parameters of marginal action distributions of $N$ agents given the input state (line 3).[2] In our implementation, we use mixture of Gaussians to realize each marginal policy $\pi_i(a_i|s; \theta_i)$ such that we can model complex multi-modal marginals while having a tractable form of the marginal cumulative distribution functions. Therefore, the output of $MLP_{marginal}$ consists of the means, covariance, and weights of components for the $N$ agents' Gaussian mixtures. We then calculate the likelihood of each observed action $a_j$ based on agent $j$'s marginal action distribution (line 5), and maximize the likelihood by optimizing the parameters of $MLP_{marginal}$ (line 6).

---

[1] An alternate approach is to combine the two steps together and use end-to-end training, but this does not perform well in practice because the copula term is unlikely to converge before marginals are well-trained.

[2] Here we assume that each agent is aware of the whole system state. But our model can be easily generalized to the case where agents are only aware of partial system state by feeding the corresponding state to their MLPs.

After learning marginals, we fix the parameters of marginal MLPs and start learning the copula (lines 7-20). We first process the original demonstrations using probability integral transform and obtain a set of new demonstrations with uniform marginals (lines 8-13). Then we learn the density of copula (lines 14-20). Notice that the copula can be implemented as either *state-dependent* (lines 14-17) or *state-independent* (lines 18-20): For state-dependent copula, we use $MLP_{copula}$ to take as input the current state $s$ and outputs the parameters of copula density $c(\cdot|s)$ (line 15). Then we calculate the likelihood of copula value $u$ (line 16) and maximize the likelihood by updating $MLP_{copula}$ (line 17). For state-independent copula, we directly calculate the likelihood of copula value $u$ under $c(\cdot)$ (line 19) and learn parameters of $c(\cdot)$ by maximizing the likelihood (line 20).

The copula density ($c(\cdot)$ or $c(\cdot|s)$) can be implemented using parametric methods such as Gaussian or mixture of Gaussians. It is worth noticing that if copula is state-independent, it can also be implemented using non-parametric methods such as kernel density estimation (Parzen, 1962; Davis et al., 2011). In this way, we no longer learn parameters of copula by maximizing likelihood as in lines 19-20, but simply store all copula values $u$ for density estimation and sampling in inference stage. We will visualize the learned copula in experiments.

**Inference and Generation.** In inference stage, the goal is to predict the joint actions of all agents given their current state $s$. The inference algorithm is presented as Algorithm 2 in Appendix A, where we first sample a copula value $u = (u_1, \cdots, u_N)$ from the learned copula, either state-dependent or state-independent (lines 1-5), then apply inverse probability transform to transform them to the original action space: $\hat{a}_j = F_j^{-1}(u_j|s)$ (lines 7-10). Note that an analytical form of the inverse cumulative distribution function may not always be available. In our implementation, we use binary search to approximately solve this problem since $F_j$ is a strictly increasing function, which is shown to be highly efficient in practice. In addition, we can also sample multiple i.i.d. copula values from $c(\cdot|s)$ or $c(\cdot)$ (line 3 or 5), transform them into the original action space, and take their average as the predicted action. This strategy is shown to be able to improve the accuracy of action prediction (in terms of MSE loss), but requires more running time as a trade-off.

The generation algorithm is presented as Algorithm 3 in Appendix A. To generate new trajectories, we repeatedly predict agent actions given the current state (line 2), then execute the generated action and obtain an updated state from the environment (line 3).

The computational complexity of the training and the inference algorithms is analyzed as follows. The complexity of each round in Algorithm 1 is $O(MTN)$, where $M$ is the number of trajectories in the training set, $T$ is the length of each trajectory, and $N$ is the number of agents. The complexity of Algorithm 2 is $O(N)$. The training and the inference algorithms scales linearly with the size of input dataset.

## 4 RELATED WORK

The key problem in multi-agent imitation learning is how to model the dependence structure among multiple interactive agents. Le et al. (2017) learn a latent coordination model for players in a cooperative game, where different players occupy different roles. However, there are many other multi-agent scenarios where agents do not cooperate for a same goal or they do not have specific roles (e.g., self-driving). Bhattacharyya et al. (2018) adopt parameter sharing trick to extend generative adversarial imitation learning to handle multi-agent problems, but it does not model the interaction of agents. Interaction Network (Battaglia et al., 2016) learns a physical simulation of objects with binary relations, and CommNet (Sukhbaatar et al., 2016) learns dynamic communication among agents. But they fail to characterize the dependence among agent actions explicitly.

Researchers also propose to infer multi-agent relationship using graph techniques or attention mechanism. For example, Kipf et al. (2018) propose to use graph neural networks (GNN) to infer the type of relationship among agents. Hoshen (2017) introduces attention mechanism into multi-agent predictive modeling. Li et al. (2020) combine generative models and attention mechanism to capture behavior generating process of multi-agent systems. These works address the problem of reasoning relationship among agents rather than capturing their dependence when agents are making decisions.

Another line of related work is deep generative models in multi-agent systems. For example, Zhan et al. (2018) propose a hierarchical framework with programmatically produced weak labels to gen-

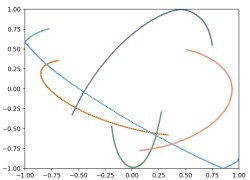 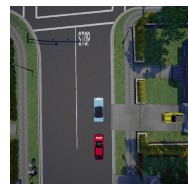 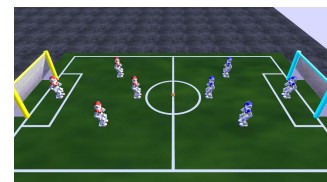

Figure 2: Experimental environments (left to right): PhySim, Driving, and RoboCup.

erate realistic multi-agent trajectories of basketball game. Yeh et al. (2019) use GNN and variational recurrent neural networks (VRNN) to design a permutation equivariant multi-agent trajectory generation model for sports games. Ivanovic et al. (2018) combine conditional variational autoencoder (CVAE) and long-short term memory networks (LSTM) to generate behavior of basketball players. Most of the existing works focus on agent behavior forecasting but provide limited information regarding the dependence among agent behaviors.

## 5 EXPERIMENTS

### 5.1 EXPERIMENTAL SETUP

**Datasets.** We evaluate our method in three settings. **PhySim** is a synthetic physical environment where 5 particles are connected by springs. **Driving** is a synthetic driving environment where one vehicle follows another along a single lane. **RoboCup** is collected from an international scientific robot competition where two robot teams (including 22 robots) compete against each other. The detailed dataset description is provided in Appendix B. Experimental environments are shown in Figure 2.

**Baselines.** We compare our method with the following baselines: **LR** is a logistic regression model that predicts actions of agents using all of their states. **SocialLSTM** (Alahi et al., 2016) predicts agent trajectory using RNNs with a social pooling layer in the hidden state of nearby agents. **IN** (Battaglia et al., 2016) predicts agent states and their interactions using deep neural networks. **CommNet** (Sukhbaatar et al., 2016) simulates the inter-agent communication by broadcasting the hidden states of all agents and then predicts their actions. **VAIN** (Hoshen, 2017) uses neural networks with attention mechanism for multi-agents modeling. **NRI** (Kipf et al., 2018) designs a graph neural network based model to learn the interaction type among multiple agents. Since most of the baselines are used for predicting future states given historical state, we change the implementation of their objective functions and use them to predict the current action of agents given historical states. Each experiment is repeated 3 times, and we report the mean and standard deviation.

### 5.2 RESULTS

We compare our method with baselines in the task of action prediction. The results of root mean squared error (RMSE) between predicted actions and ground-truth actions are presented in Table 1. The number of Gaussian mixture com-

|  | LR | SocialLSTM | IN | CommNet | VAIN | NRI | Copula |
|---|---|---|---|---|---|---|---|
| **PhySim** | 0.064 ± 0.002 | 0.186 ± 0.032 | 0.087 ± 0.013 | 0.089 ± 0.007 | 0.082 ± 0.010 | 0.055 ± 0.011 | **0.037** ± 0.005 |
| **Driving** | 0.335 ± 0.007 | 0.283 ± 0.024 | 0.247 ± 0.033 | 0.258 ± 0.028 | 0.242 ± 0.031 | 0.296 ± 0.018 | **0.158** ± 0.019 |
| **RoboCup** | 0.478 ± 0.009 | 0.335 ± 0.051 | 0.320 ± 0.024 | 0.311 ± 0.042 | 0.315 ± 0.028 | 0.401 ± 0.042 | **0.221** ± 0.024 |

Table 1: Root mean squared error (RMSE) between predicted actions and ground-truth actions for our method and baselines.

ponents in our method is set to 2 for all datasets. The results demonstrate that all methods performs the best on PhySim dataset, since agents in PhySim follow simple physical rules and the relationships among them are linear thus easy to infer. However, the interactions of agents in Driving and RoboCup datasets are more complicated, which causes LR and NRI to underperform other baselines. The performance of IN, CommNet, and VAIN are similar, which is in accordance with the result reported in Hoshen (2017). Our method is shown to outperform all baselines significantly

on all three datasets, which demonstrates that explicitly characterizing dependence of agent actions could greatly improve the performance of multi-agent behavior modeling.

To investigate the efficacy of copula, we implement three types of copula function: Uniform copula means we do not model dependence among agent actions. KDE copula uses kernel density estimation to model the copula function, which is state-independent. Gaussian mixtures copula uses Gaussian mixture model to characterize the copula function, of which the parameters are output by an MLP taking as input the current state. We train the three models on training trajectories, then calculate negative log-likelihood (NLL) of test

|  | Uniform copula | KDE copula | Gaussian mix. copula |
|---|---|---|---|
| **PhySim** | $8.994 \pm 0.001$ | $\mathbf{1.256} \pm 0.006$ | $2.893 \pm 0.012$ |
| **Driving** | $-0.571 \pm 0.024$ | $\mathbf{-0.916} \pm 0.017$ | $-0.621 \pm 0.028$ |
| **RoboCup** | $3.243 \pm 0.049$ | $\mathbf{0.068} \pm 0.052$ | $3.124 \pm 0.061$ |

Table 2: Negative log-likelihood (NLL) of test trajectories evaluated by different copula. Uniform copula assumes no dependence among agent actions. KDE copula uses kernel density estimation to model the copula, which is state-independent. Gaussian mixtures copula uses Gaussian mixture model to characterize the copula, which is state-dependent.

trajectories using the three trained models. A lower NLL score means that the model assigns high likelihood to given trajectories, showing that it is better at characterizing the dataset. The NLL scores of the three models on the three datasets are reported in Table 2. The performance of KDE copula and Gaussian copula both surpasses uniform copula, which demonstrates that modeling dependence among agent actions is essential for improving model expressiveness. However, Gaussian copula performs worse than KDE copula, because Gaussian copula is state-dependent thus increases the risk of overfitting. Notice that the performance gap between KDE and Gaussian copula is less on PhySim, since PhySim dataset is much larger so the Gaussian copula can be trained more effectively.

## 5.3 GENERALIZATION CAPABILITY OF COPULA

One benefit of copulas is that copula captures the pure dependence among agents, regardless of their own marginal action distributions. To demonstrate the generalization capabilities of copulas, we design the following experiment. We first train our model on

|  | Original marginals + original copula | Original marginals + new copula | New marginals + original copula | New marginals + new copula |
|---|---|---|---|---|
| **PhySim** | $10.231 \pm 0.562$ | $8.775 \pm 0.497$ | $1.301 \pm 0.016$ | $1.259 \pm 0.065$ |
| **Driving** | $15.184 \pm 1.527$ | $13.662 \pm 0.945$ | $0.447 \pm 0.085$ | $-0.953 \pm 0.024$ |
| **RoboCup** | $4.278 \pm 0.452$ | $4.121 \pm 0.658$ | $0.114 \pm 0.020$ | $0.077 \pm 0.044$ |

Table 3: Negative log-likelihood (NLL) of new test trajectories in which the action distribution of one agent is changed. We evaluate the new test trajectories based on whether to use the original marginal action distributions or copula, which results in four combinations.

the original dataset, and learn marginal action distributions and copula function (which is called *original marginals* and *original copula*). Then we substitute one of the agents with a new agent and use the simulator to generate a new set of trajectories. Specifically, this is achieved by doubling the action value of one agent (for example, this can be seen as substituting an existing particle with a lighter one in PhySim). We retrain our model on new trajectories and learn *new marginals* and *new copula*. We evaluate the likelihood of new trajectories based on whether to use the original marginals or original copula, which, accordingly, results in four combinations. The NLL scores of four combinations are presented in Table 3. It is clear, by comparing the first and the last column, that "new marginals + new copula" significantly outperform "original marginals + original copula", since new marginals and new copula are trained on new trajectories and therefore characterize the new joint distribution exactly. To see the influence of marginals and copula more clearly, we further compare the results in column 2 and 3, where we use new copula or new marginals separately. It is clear that the model performance does not drop significantly if we use the original copula and new marginals (by comparing column 3 and 4), which demonstrates that the copula function basically stays the same even if marginals are changed. The result supports our claim that the learned copula is generalizable in the case where marginal action distributions of agents change but the internal inter-agent relationship stays the same.

## 5.4 COPULA VISUALIZATION

Another benefit of copulas is that it is able to intuitively demonstrate the correlation among agent actions. We choose the RoboCup dataset to visualize the learned copula. As shown in Figure 4a

in Appendix D, we first randomly select a game (the 6th game) between cyrus2017 and helios2017 and draw trajectories of 10 players in the left team (L2 $\sim$ L11, except the goalkeeper). It is clear that the 10 players fulfill specific roles: L2 $\sim$ L4 are defenders, L5 $\sim$ L8 are midfielders, and L9 $\sim$ L11 are forwards. Then we plot the copula density between the x-axis (the horizontal direction) of L2 and the x-axis of L3 $\sim$ L11, respectively, as shown in Figure 4b in Appendix D. These figures illustrate linear correlation between their moving direction along x-axis, that is, when L2 moves forward other players are also likely to move forward. However, the correlation strength differs with respect to different players according to the visualized result: L2 exhibits high correlation with L3 and L4, but low correlation with L9 $\sim$ L11. This is because L2 $\sim$ L4 are all defenders so they collaborate more closely with each other, but L9 $\sim$ L11 are forwards thus far from L2 in the field.

## 5.5 TRAJECTORY GENERATION

The learned copula can also be used to generate new trajectories. We visualize the result of trajectory generation on RobuCup dataset. As shown in Figure 3, the dotted lines denote the ground-truth trajectories of the 10 player in an attack from midfield to the penalty area. The trajectories generated by our copula model (Figure 3b) are quite similar to the demonstration as they exhibit high consistency. It is clear that midfielders and forwards (No. 5 $\sim$ No. 11) are basically moving to the same direction, and they all make a left turn on their way to penalty area. However, the generated trajectories by independent modeling show little corre-

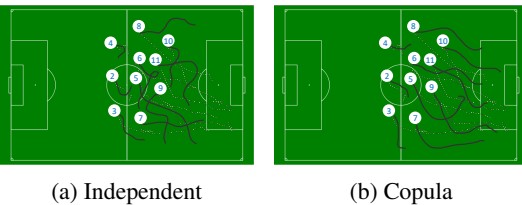

(a) Independent   (b) Copula

Figure 3: Generated trajectories on RoboCup dataset using independent modeling or copula. Dotted lines are ground-truth trajectories and solid lines are generated trajectories.

lation since the players are all making independent decisions. We also present the result of trajectory generation on Driving dataset in Appendix E.

## 6 CONCLUSION AND FUTURE WORK

In this paper, we propose a copula-based multi-agent imitation learning algorithm that is interpretable, efficient and scalable to model complex multi-agent interactions. Sklar's theorem allows us to separately learn marginal policies that capture the local behavioral patterns of each individual agent and a copula function that only and fully captures the dependence structure among the agents. Compared to previous multi-agent imitation learning methods based on independent policies (mean-field factorization of the joint policy) or opponent modeling, our method is capable of modeling complex dependence among agents and achieving coordination without any modeling redundancy. Experimental results on physical simulation, driving and robot soccer datasets demonstrate the effectiveness of our method compared with state-of-the-art baselines.

We point out two directions of future work. First, the copula function is generalizable only if the dependence structure of agents (i.e., their role assignment) is unchanged. Therefore, it is interesting to study how to efficiently apply the learned copula to the scenario with evolving dependence structure. Another practical question is that whether our proposed method can be extended to the setting of decentralized execution, since the step of copula sampling (line 3 or 5 in Algorithm 2) is shared by all agents. A straightforward way to solve this problem is to set a fixed sequence of random seeds for all agents in advance, so that the copula sample obtained by all agents is the same at each timestamp. Designing a more robust and elegant mechanism for decentralized execution is also a promising direction.

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

## A    PSEUDO CODE FOR TRAINING, INFERENCE, AND GENERATION PROCEDURE

The pseudo code for inference and generation procedure are presented in Algorithm 2 and 3, respectively.

---

**Algorithm 2:** Inference procedure

---
**Input:** Marginal action distribution MLP $MLP_{marginal}$, state-dependent copula MLP $MLP_{copula}$ or state-independent copula density $c(\cdot)$, current state $s$

**Output:** Predicted action $\hat{a}$

    // Sample from copula

1  **if** *copula is set as state-dependent* **then**

2     Calculate (parameters of) state-dependent copula density $c(\cdot|s) \leftarrow MLP_{copula}(s)$

3     Sample a copula value $u = (u_1, \cdots, u_N)$ from $c(\cdot|s)$

4  **else**

5     Sample a copula value $u = (u_1, \cdots, u_N)$ from $c(\cdot)$

    // Transform from copula space to action space

6  Calculate (parameters of) the conditional marginal action distributions for all agents:

    $\{f_j(\cdot|s)\}_{j=1}^N \leftarrow MLP_{marginal}(s)$

7  **for** *agent $j = 1, \cdots, N$* **do**

8     $F_j(\cdot|s) \leftarrow$ CDF of $f_j(\cdot|s)$

9     $\hat{a}_j \leftarrow F_j^{-1}(u_j|s)$

10  $\hat{a} \leftarrow (\hat{a}_1, \cdots, \hat{a}_j)$

11  **return** $\hat{a}$

---

 

---

**Algorithm 3:** Generation procedure

---
**Input:** Inference module (Algorithm 2), initial state $s[0]$, required length $L$, environment $\mathcal{E}$

**Output:** Generated trajectory $\hat{\tau}$

1  **for** $l = 0, \cdots, L$ **do**

2     Feed state $s[l]$ to the inference module and get the predicted action $\hat{a}[l]$

3     Execute $\hat{a}[l]$ in environment $\mathcal{E}$ and get a new state $s[l+1]$

4  $\hat{\tau} = \{(s[l], \hat{a}[l])\}_{l=0}^L$

5  **return** $\hat{\tau}$

---

## B    DATASET DETAILS

**PhySim** is collected from a physical simulation environment where 5 particles move in a unit 2D box. The state is locations of all particles and the action is their acceleration (there is no need to include their velocities in state because accelerations are completely determined by particle locations). We add Gaussian noise to the observed values of actions. Particles may be pairwise connected by springs, which can be represented as a binary adjacency matrix $\mathbf{A} \in \{0, 1\}^{N \times N}$. The elasticity between two particles scales linearly with their distance. At each timestamp, we randomly sample an adjacency matrix from $\{\mathbf{A}_1, \mathbf{A}_2\}$ to connect all particles, where $\mathbf{A}_1$ and $\mathbf{A}_2$ are set as complimentary (i.e. $\mathbf{A}_1 + \mathbf{A}_2 + \mathbf{I} = \mathbf{1}$) to ensure that they are as different as possible. Therefore, the marginal action distribution of each particle given a system state is Gaussian mixtures with two components. Here the coordination signal for particles can be seen as the hidden variable determining which set of springs ($\mathbf{A}_1$ or $\mathbf{A}_2$) is used at current time. We generate $10,000$ training trajectories, $2,000$ validation trajectories, and $2,000$ test trajectories for experiments, where the length of each trajectory is 500.

**Driving** is generated by CARLA[3] (Dosovitskiy et al., 2017), an open-source simulator for autonomous driving research that provides realistic urban environments for training and validation of autonomous driving systems. To generate the driving data, we design a car following scenario,

---

[3]https://carla.org/

where a leader car and a follower car drive in the same lane. We make the leader car alternatively accelerate to a speed upper bound and slow down to stopping. The leader car does not care about the follower and drives following its own policy. The follower car tries to follow closely the leader car while keeping a safe distance. Here the state is the locations and velocities of the two cars, and the action is their accelerations. We generate $1,009$ trajectories in total, and split the whole data into training, validation, and test set with ratio of $6:2:2$. The average length of trajectories is $85.5$ in Driving dataset.

**RoboCup** (Michael et al., 2017) is collected from an international scientific robot football competition in which teams of multiple robots compete against each other. The original dataset contains all pairings of 10 teams with 25 repetitions of each game ($1,125$ games in total). The state of a game (locations and velocities of 22 robots) is recorded every $100$ ms, resulting in a trajectory of length $6,000$ for each game (10 min). We select the 25 games between two teams, cyrus2017 and helios2017, as the data used in this paper. The state is locations of 10 robots (except the goalkeeper) in the left team, and the action is their velocities. The dataset is split into training, validation, and test set with ratio of $6:2:2$.

For the three datasets, to learn the marginal action distribution of each agent (i.e. Gaussian mixtures), we use an MLP with one hidden layer to take as input a state and output the centers of their Gaussian mixtures. To prevent overfitting, the variance of these Gaussian mixtures is parameterized by a free variable for each particle, and the weights of mixtures are set as uniform. Each dimension of states and actions in the original datasets are normalized to range $[-1, 1]$. For PhySim, the number of particles are set to $5$. Learning rate is set to $0.01$, and the weight of L2 regularizer is set to $10^{-5}$. For Driving, learning rate is $0.005$ and L2 regularizer weight is $10^{-5}$. For RoboCup, learning rate is $0.001$ and L2 regularizer weight is $10^{-6}$.

## C  BASELINE IMPLEMENTATION DETAILS

For LR, we use the default implementation in Python sklearn package. For SocialLSTM (Alahi et al., 2016), the dimension of input is set as the dimension of states in each dataset. The spatial pooling size is 32, and we use an $8 \times 8$ sum pooling window size without overlaps. The hidden state dimension in LSTM is $128$. The learning rate is $0.001$. For IN (Battaglia et al., 2016), all MLPs are with one hidden layer of 32 units. The learning rate is $0.005$. For CommNet (Sukhbaatar et al., 2016), all MLPs are with one hidden layer of 32 units. The dimension of hidden states is set to $64$, and the number of communication round is set to $2$. The learning rate is $0.001$. For VAIN (Hoshen, 2017), the encoder and decoder functions are implemented as a fully connected neural network with one hidden layer of 32 units. The dimension of hidden states is $64$, and the dimension of attention vectors is $10$. The learning rate is $0.0005$. For **NRI** (Kipf et al., 2018), we use an MLP encoder and an MLP decoder, with one hidden layer of 32 units. The learning rate is $0.001$.

## D  VISUALIZED COPULA ON ROBOCUP DATASET

The trajectories of players in one game as well as the visualized pairwise copula are presented in Figure 4.

## E  GENERATED TRAJECTORIES ON DRIVING DATASET

For the Driving dataset, we randomly select 10 original trajectories and 10 trajectories generated by our method, and show the visualization results in Figure 5. The x-axis is timestamp and y-axis is the location (coordinate) of two cars. Our learned policy is shown to be able to maintain the distance between two cars.

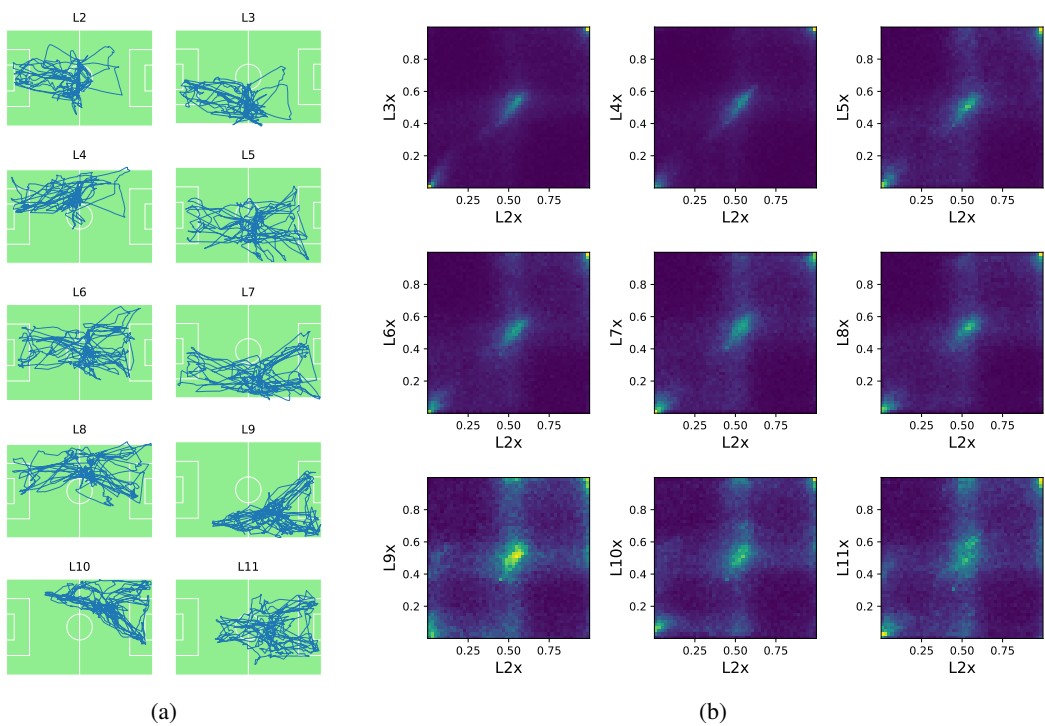

Figure 4: (a) Trajectories of 10 players (except the goalkeeper) of the left team in one RoboCup game; (b) Copula density between x-axis of the L2 player and x-axis of another player (L3 ∼ L11).

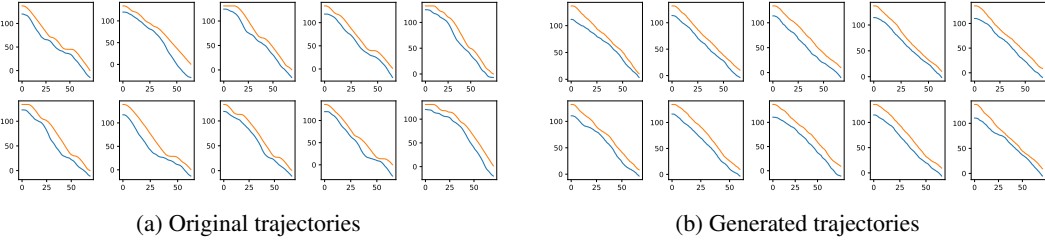

Figure 5: Original and generated trajectories on Driving dataset. The x-axis is timestamp and y-axis is the location (1D coordinate) of two cars.

