# OpenReview forum: "Multi-Agent Imitation Learning with Copulas"
_ICLR.cc/2021/Conference — Reject_

### Official Review · AnonReviewer3 · 2020-10-28
**Interesting problem but does the proposed solution really help?**

**Rating:** 4
**Confidence:** 5

**Review:**

This paper discusses the use of copulas for the multi-agent imitation learning scenario. The paper addresses an interesting application of copulas and is generally well written. Efficient imitation learning for MAS is a challenging problem, hence the paper is well motivated. While I like the overall idea of the paper, there are some concerns which the authors should address:

1. The authors claim that learning a copula is more efficient and scalable, however I don't feel this claim is well justified in the paper. In general, a copula captures arbitrary dependencies between the action distributions of $n$ agents, how exactly is then learning a copula any different from learning a joint distribution $\pi(a_1..a_n)$ in terms of learning complexity?

2. The authors should mention the approximate order complexity of the Algorithms 1, 2 and in general need to compare those with the competing methods mentioned in the paper to give a better understanding of the advantage using copula based policy learning.

3.  In general, maximising over a copula seems like a very hard problem, which is one of the reasons why mean-field methods are proposed, how exactly is the copula based factorisation avoiding this problem?

4. The authors should provide the architecture details of baseline methods to better interpret the results.

5. Root mean squared error (RMSE) between predicted actions and ground-truth actions for Table 1 exps is not a very convincing metric, specially from the standpoint of qualitative nature of policies imitated. It would be better if the authors can report reward scores for some suitable domains (ex. RoboCup).

---

> ### Author Response · Authors · 2020-11-13
> **Authors' Response to Reviewer 3**
>
> We appreciate the reviewer’s helpful and detailed feedback. We have addressed your comments and included further clarifications in the paper. The updates in our paper are marked in red.
>
> Q1: *The authors claim that learning a copula is more efficient and scalable, however I don't feel this claim is well justified in the paper. In general, a copula captures arbitrary dependencies between the action distributions of n agents, how exactly is then learning a copula any different from learning a joint distribution π(a1..an) in terms of learning complexity?*
>
> A1: In terms of learning complexity, our copula-based policy parametrization is indeed the same as directly learning a joint distribution. However, we want to clarify that it is more efficient and scalable in transfer learning scenarios. To begin with, as shown in Figure 1, two different joint distributions may share the same copula (dependency structure) while having different marginals, or share the marginals while having different copulas. In some scenarios where the marginal policy of some agent changes but the dependency among the agents remains the same (e.g., in a soccer game, one player is replaced by his/her substitute, but the dependence among different roles are basically the same regardless of players), we can obtain a new joint policy efficiently by switching in the new agent’s marginal while keeping the copula and other marginals unchanged. If we directly use a function approximator (e.g. neural networks) to learn the joint policy for the original team, then even if only a single marginal is changed, we need to re-learn the whole joint distribution. In contrast, using copula-based policy, we only need to learn the changed marginal while keeping the other components unchanged. Thus our method offers more flexibility and scalability in these scenarios. Moreover, the experiments in section 5.3 empirically demonstrate these benefits.
>
>
> Q2: *What is the approximate order complexity of the Algorithms 1 and 2?*
>
> A2: We have added the complexity analysis in the updated paper (page 6). The proposed algorithm scales linearly with the input dataset, which is superior to most baselines which is with $O(N^2)$ complexity where $N$ is the number of agents.
>
>
> Q3: *In general, maximizing over a copula seems like a very hard problem, which is one of the reasons why mean-field methods are proposed, how exactly is the copula based factorization avoiding this problem?*
>
> A3: We think that modeling the dependency structure among agents is the central problem for multi-agent learning tasks. In our context, such dependency structure is encoded in the copula. The central contribution of our paper is a practical framework that enables us to approximately learn a copula-based policy. As for the exact proposed solution (steps for learning the copula and sampling from the copula-based policy), please refer to section 3.3 as well as appendix A, where we have included many training/inference details as well as pseudo code. Basically, for the training phase, we first learn marginals (single-agent imitation learning problem), then process the data with the corresponding CDFs, then learn the copula function based on the data obtained by probability integral transform. For the sampling phase, we first sample random variables from the copula distribution, then apply inverse probability integral transform to obtain action samples.
>
>
> Q4: *What are the architecture details of baseline methods?*
>
> A4: We have added the baseline implementation details in Appendix C (page 13).
>
>
> Q5: *Root mean squared error (RMSE) between predicted actions and ground-truth actions for Table 1 exps is not a very convincing metric, especially from the standpoint of qualitative nature of policies imitated. It would be better if the authors can report reward scores for some suitable domains (ex. RoboCup).*
>
> A5: Previous work (e.g., [VAIN](https://proceedings.neurips.cc/paper/2017/file/748ba69d3e8d1af87f84fee909eef339-Paper.pdf)) also uses RMSE as a metric to evaluate the quality of prediction. We use RMSE here because the actions lie in a continuous space. Moreover, we also use log-likelihood as an additional metric. Reward may not be a good metric, because we may learn a policy that is quite different from the target policy but obtains good reward. For example, the expert policy generates diverse trajectories while a learned policy may be able to generate only a single trajectory with the highest reward. In imitation learning, the goal is to obtain a parametric policy that exactly matches the target policy.

---

> ### Author Response · Authors · 2020-11-23
> **Rebuttal follow-up**
>
> We think we have addressed  your questions and concerns in the rebuttal and we would really appreciate your feedback. Please let us know if there is some other information we can provide to assist in the evaluation of our paper. Thanks a lot for your time!

---

### Official Review · AnonReviewer2 · 2020-10-28

**Rating:** 5
**Confidence:** 3

**Review:**

The paper proposes a multi-agent imitation learning method that learns a joint policy for all agents from offline demonstrations. The key idea is to first learn a marginal policy for each agent using behavioral cloning, then fit a copula function that captures dependencies between the agents' policies. Experiments with particle simulations, simulated driving, and simulated RoboCup suggest that this approach could potentially outperform prior methods.

Theorem 1 seems like a powerful tool for multi-agent imitation learning, and I think this paper is a promising initial step. That being said, there are several issues:

In Tables 2 and 3, there are negative values for the negative log-likelihood. This is not possible, since log-likelihoods cannot be positive. I initially assumed that this was a typo in Table 2, but then I saw that Table 3 included both negative and positive values for the NLL, which makes me unsure about the overall correctness of the experimental evaluations in this paper.

Algorithm 1 uses behavioral cloning to fit the marginal policies. Doesn't this lead to compounding errors due to state distribution mismatch? Would more recent methods for offline imitation learning that overcome this issue, like ValueDICE [1], be a more suitable choice than behavioral cloning?

Doesn't learning the marginals first, then the copula second, lead to a model misspecification when fitting the marginals? Learning the marginals separately (without a copula) effectively assumes the factorization in Equation 1, which we know cannot represent the ground-truth joint policies in the experiments, so the learned marginals from step 1 are not necessarily the same marginals that you would learn if you were to fit the marginals and the copula simultaneously.

From Equation 7, we can see that different combinations of marginals and copulas could potentially induce the same joint policy. If you only fit the model by maximizing the log-likelihood of demonstrations for a single task, how can you distinguish between different pairs of (marginals, copulas) that fit the demonstrations equally well? Wouldn't you need to train on multiple tasks that share the same underlying copula but have different marginals, or on multiple tasks that share the same underlying marginals but have different copulas?

It would be nice to discuss the weaknesses of the proposed method, as well as directions for future work. For example, could this approach to learning a joint policy be extended to settings that require decentralized execution?

The dotted lines in Figure 3 are very difficult to see because they are small and because they are dark gray against a dark green background.

It would be nice to include the pseudocode in the main paper, since the main contribution of the paper is the algorithm.

[1] https://openreview.net/pdf?id=Hyg-JC4FDr

Update after rebuttal
-----------------------------
Thanks to the authors for addressing my concerns. I have updated my score.

---

> ### Author Response · Authors · 2020-11-13
> **Authors' Response to Reviewer 2**
>
> We really appreciate the reviewer’s helpful and detailed feedback. We have answered the questions below, and the updates in our paper are marked in red.
>
> Q1: *The negative log-likelihood can not be negative values in Tables 2 and 3.*
>
> A1: Actually, negative log-likelihood can be negative. This is because our action space is continuous, so the likelihood of an action (i.e., $p(x)$ where $p()$ is a probability **density** function and $x$ is an action) can be greater than 1. This is different from the case of a discrete action space, where the likelihood cannot exceed 1. Please refer to [here](https://stats.stackexchange.com/questions/140463/can-the-likelihood-take-values-outside-of-the-range-0-1#:~:text=2%20Answers&text=Likelihood%20must%20be%20at%20least,can%20be%20greater%20than%201) for more information.
>
> Q2: *Doesn't imitation learning lead to compounding errors due to state distribution mismatch? Would more recent methods for offline imitation learning that overcome this issue be a more suitable choice than behavioral cloning?*
>
> A2: Yes, in terms of learning marginal policies ($\pi_i(a_i|s)$) (single-agent imitation learning problem), there are potentially better methods but they are **completely compatible** with our frameworks. We want to clarify that, the **central contribution** of our paper is to integrate the copula-based policy factorization into multi-agent imitation learning and demonstrate various benefits brought by our new framework. Recall that our framework involves learning marginals for individual agents as well as a copula. Thus it is indeed promising to combine more recent offline imitation learning methods with our advances for learning better marginals and we leave them as exciting future directions.
>
> Q3: *Doesn't learning the marginals first, then the copula second, lead to a model misspecification when fitting the marginals? Learning the marginals separately (without a copula) effectively assumes the factorization in Equation 1, which we know cannot represent the ground-truth joint policies in the experiments, so the learned marginals from step 1 are not necessarily the same marginals that you would learn if you were to fit the marginals and the copula simultaneously.*
>
> A3: Actually, given expressive enough functions (if not, any model is misspecified without assumptions on the data generating process), “learning the marginals first, then the copula second” can approximate any distribution arbitrarily well. From Theorem 1 (also see Eq. 6 and 7), any joint distribution can be factorized as the product of marginals and a copula. No matter how complex the joint distribution $p(a_1, \ldots, a_N)$ is, it must induce $N$ marginal policies and the copula only and fully describes the dependency structure left after we remove the marginal information by processing the data with the learned marginal CDF (via probability integral transform). Hence, by the Sklar’s theorem, our policy factorization and optimization is general and will not lead to model misspecification.
>
> Q4: *Different combinations of marginals and copulas could potentially induce the same joint policy. How can you distinguish between different pairs of (marginals, copulas) that fit the demonstrations equally well?*
>
> A4: It is true that different combinations of marginals and copulas could potentially induce the same joint distribution, but there is no need to consider all these combinations. Recall that the goal of imitation learning is, given i.i.d. data sampled from some unknown target distribution, learn a parametric distribution that approximates the target distribution. As long as our algorithm can find one combination that fits the data distribution reasonably well, it is a good solution. Furthermore, theoretically under mild assumptions (marginal CDFs being continuous), the factorization in theorem 1 is unique, which means the optimal solution is also unique.
>
> Q5: *The weaknesses of the proposed method and future work. For example, could this approach to learning a joint policy be extended to settings that require decentralized execution?*
>
> A5: We have revised the last section of our paper and include the weaknesses and future work (page 9). Regarding the question raised by the reviewer here: Yes. Since modern computer systems rely on pseudorandom number generator, a straightforward way for decentralized execution of the proposed method is to distribute a fixed sequence of random seeds for all agents in advance, then each agent uses the same random seed sequence to sample from copula density and calculate their actions with the learned marginals.
>
> Q6: *Figure 3 is unclear.*
>
> A6: We have updated Figure 3 to make the dotted lines more clear (page 9).
>
> Q7: *It would be nice to include the pseudocode in the main paper, since the main contribution of the paper is the algorithm.*
>
> A7: We have revised the paper and put Algorithm 2 back to the main paper (page 5). We will also make our implementation publicly available.

---

> ### Author Response · Authors · 2020-11-23
> **Rebuttal follow-up**
>
> Thanks a lot for reading our rebuttal and updating the score! We would like to know whether there are other concerns left and we would really appreciate feedbacks on how to improve the paper further. Thanks again for your time!

---

### Official Review · AnonReviewer5 · 2020-11-09
**Learning a joint representation of agent policies in multi agent games**

**Rating:** 7
**Confidence:** 3

**Review:**

Summary:

The paper proposes the use of "Copulas" to capture dependencies among agents in multi-player environments. The authors argue that prior work(eg GNN, VAE, LSTM etc based) do not leverage the common structure among agent behaviour. They explain how the copula can more efficiently encode the dependency among policies, as compared to simply factorizing. The authors provide empirical results on real world and synthetic datasets, showcasing superior performance against baseline approaches.

Overall, this seems like a novel approach with promising results on a few experimental set ups. The authors describe how their proposed model builds upon the limitations of prior work. The work seems moderately significant.

Comments:

* The writing is easy to follow. The paper motivate the problem well and describe the experimentation clearly.

* Nit: In Figure 1b) Can the authors clarify how the copula's differ in the upper figure? It seems to be a uniform distribution, I assume this applies for only the lower figure.

* For equation (3), can the notation be clarified: this refers to learning opponent policies rather than agents co-operating in a given task?

* Nit: Algorithm 2 can perhaps be described in the main section, being the central idea to this paper.

* For the datasets, can the number of agents in each be briefly described in Section 5.1 or Table 1? Is there any dependence on performance with respect to this parameter? The authors have mentioned heir intuition behind some of the results in Section 5.2.

* In Section 5.3, how were the replacement agents sampled? For eg, in the RoboCup dataset used, would substituting a defender for a forward influence the performance(as hinted in Section 5.4)?


Given the above points, novelty and significance  marking this as accept.

Update:
After seeing the author response below, no change to my score.

---

> ### Author Response · Authors · 2020-11-13
> **Authors' Response to Reviewer 5**
>
> We thank the reviewer for their helpful and detailed feedback. Reviewer makes a number of helpful suggestions, and we have addressed your comments and included further clarifications in the paper. The updates in our paper are marked in red.
>
> Q1: *The meaning of Figure 1b.*
>
> A1: The copula density in the upper part of Figure 1b is uniform in the space of $[0,1]^2$, and the copula density in the lower part of Figure 1b is uniform over $[0, \frac{1}{2}]^2 \cup [\frac{1}{2},1]^2$. We have revised the description in Figure 1b to make this clear.
>
> Q2: *For equation 3, does this refer to learning opponent policies rather than agents cooperating in a given task?*
>
> A2: In this paper, we do not need to make assumptions on whether the task is cooperative or competitive. The joint policy can always be decomposed as Eq (3) regardless of the task type.
>
>
> Q3: *Algorithm 2 can perhaps be described in the main section.*
>
> A3: Yes, we agree. Due to the page limit, we did not put Algorithm 2 in the main section. We have revised the paper and put Algorithm 2 back to the main paper (page 5).
>
>
> Q4: *What is the number of agents for datasets? What is the impact of the number of agents on performance?*
>
> A4: We have revised Section 5.1 and added the number of agents for all datasets (page 7). We changed the number of agents for PhySim dataset from 2 to 10, and our method is shown to consistently outperform baselines regardless of the number of agents.
>
>
> Q5: *How are agents replaced? Would replacing a forward by a defender influence the performance?*
>
> A5: In the experiment of agent replacement, we first randomly select an agent from the system. Suppose the marginal action distribution of this agent is $f(a|s)$, and his action is generated by $a_{old} \sim f(a|s)$ given a state $s$. Then this agent is replaced by a new agent whose action is $a_{new} = 2 a_{old}$, where $a_{old}$ is sampled from $f(a|s)$ that is the same as before. This means that the agent will be faster (since its action value is twice as much as before), but its role (defender or forward) is not changed. If a forward is indeed replaced by a defender, the dependency structure among agent actions will be changed, therefore the copula function is no longer generalizable to the new case.

---

### Decision · Program_Chairs · 2021-01-07
**Final Decision**

**Decision:**

Reject

**Comment:**

This paper applies an existing tool (copulas) to MARL to represent dependencies between variables.

The reviewers appreciate the use of the copulas for this problem. The experimental section shows promising results on several problems. I appreciate that the authors have answered and addressed many points of concerns of the reviewers. The paper is well written

The reviewers seem to see this paper as a first step only, showing promising results but of moderate significance in itself. In particular, reviewer 3 would like to see more justifications for the use of the copulas, and a more experimental settings would make a stronger paper.